# Human-to-Robot Handover Based on Reinforcement Learning

**DOI:** 10.3390/s24196275

**Published:** 2024-09-27

**Authors:** Myunghyun Kim, Sungwoo Yang, Beomjoon Kim, Jinyeob Kim, Donghan Kim

**Affiliations:** 1Department of Electrical Engineering (Age Service-Tech), Kyung Hee University, Seoul 02447, Republic of Korea; kmh8667@khu.ac.kr (M.K.); p1112007@khu.ac.kr (S.Y.); 2Department of Artificial Intelligence, College of Software, Kyung Hee University, Seoul 02447, Republic of Korea; 1222kbj@khu.ac.kr (B.K.); wls2074@khu.ac.kr (J.K.)

**Keywords:** reinforcement learning, manipulator, anthropomorphic gripper, handover

## Abstract

This study explores manipulator control using reinforcement learning, specifically targeting anthropomorphic gripper-equipped robots, with the objective of enhancing the robots’ ability to safely exchange diverse objects with humans during human–robot interactions (HRIs). The study integrates an adaptive HRI hand for versatile grasping and incorporates image recognition for efficient object identification and precise coordinate estimation. A tailored reinforcement-learning environment enables the robot to dynamically adapt to diverse scenarios. The effectiveness of this approach is validated through simulations and real-world applications. The HRI hand’s adaptability ensures seamless interactions, while image recognition enhances cognitive capabilities. The reinforcement-learning framework enables the robot to learn and refine skills, demonstrated through successful navigation and manipulation in various scenarios. The transition from simulations to real-world applications affirms the practicality of the proposed system, showcasing its robustness and potential for integration into practical robotic platforms. This study contributes to advancing intelligent and adaptable robotic systems for safe and dynamic HRIs.

## 1. Introduction

In recent years, with the advancement of artificial intelligence (AI) and the accelerated development of robots, the development of collaborative robots that can work with humans has increased, extending beyond industrial settings to include manipulators that provide services in daily life. A gripper has various designs, including two-finger designs and those tailored for specific tasks [1]. Many manipulator robots currently deployed in service handle tasks such as cooking food and making drinks. However, challenges still exist when it comes to tasks involving the exchange of items between humans and robots.

Most traditional manipulator control methods use an inverse kinematics solution to guide the end-effector to the target position, improving methods such as the transposed Jacobian approach using a conditioned mass matrix [2]. These methods are precise in controlling posture through computation, making them suitable for repetitive tasks at specific locations. However, they lack adaptability in dynamic and uncertain environments, making it challenging to handle situations such as obstacles or changes in object shape.

When the gripper attached to the end-effector is an anthropomorphic gripper rather than a simple shape like a two-finger gripper, modeling becomes more complex. A two-finger gripper requires only two contact points with the object, allowing more options for gripping, but it is less stable than an anthropomorphic gripper, especially when handling cylindrical or elongated objects where torque may be generated. Therefore, for human–robot interaction, an anthropomorphic gripper that can handle various objects and provide higher grip stability is preferable. However, the anthropomorphic gripper is more complex than the two-finger gripper because it has more fingers and includes a thumb, making it challenging to determine a grasping posture. As mentioned earlier, traditional methods require a long time for modeling and need modification whenever conditions change.

On the other hand, reinforcement learning offers advantages over traditional control methods, such as requiring less time to build a controller and having better adaptability to complex environments.

Reinforcement learning is based on an agent interacting with its environment in order to learn how to make the best decisions for the next space [3]. It involves trial and error in various scenarios to discover optimal actions. Consequently, manipulator robots employing reinforcement learning adaptively determine their subsequent actions, aiming to maximize rewards through real-time interactions with the environment. This approach excels in its ability to handle diverse variables and adapt to dynamic environments. Therefore, this study proposes a reinforcement learning controller that enables a manipulator equipped with an anthropomorphic gripper to grasp objects handed over randomly by a human.

In the field of human–robot interactions (HRIs), research on manipulators focuses on various technologies that facilitate object handovers between humans and robots. These studies encompass areas such as human–robot communication [4,5,6], grasp planning [7,8,9], human recognition, handover [10], and gripper force control [11]. These technologies are essential for ensuring successful handovers. Therefore, this paper includes object recognition when a human hands over an object, determination of the grasping point, and control of the manipulator. The force control of the gripper according to the object uses the HRI hand, an open-source anthropomorphic gripper developed in a previous study [1]. Specifically, object recognition and coordinate calculations are performed using YOLO v3 (You Only Look Once) [12] and object tracker packages [13]. The manipulator’s posture control is conducted using the PPO algorithm for reinforcement learning. The rest of the paper is organized as follows: Section 2 reviews existing research on manipulator and gripper control for handovers. Section 3 details the agent and environment setup for reinforcement learning. Section 4 describes the experiments conducted based on reward mechanisms and object inclinations and applies these results to actual robotic systems. Finally, Section 5 presents the main conclusions and limitations of the study.

## 2. Related Work

In HRIs, the process of handling objects necessitates a multitude of technologies. Several studies have focused on enhancing handover stability. Taunyazov et al. [14] developed a system that integrates sensors into robot grippers to detect the texture of objects. This system accurately identifies the characteristics of objects and contributes to their safe handling by robots during their interactions with humans. Another study by Pang et al. [15] enabled safe handovers between humans and robots using a technique that combined recorded video and cognitive algorithms. They utilized robot vision to recognize and handle unknown objects precisely, thereby enhancing the safety and efficiency of robot handovers. Christen et al. [16] studied the process of robots learning to receive objects from humans using reinforcement learning and point-cloud data. Their research focused on motion and grasp planning, as well as the optimization of robot movements, to implement a more natural handover experience.

In handovers, the movement of the gripper is as important as that of the manipulator because the action of grasping an object is essential for safe reception. Similarly, a study by Wang et al. [17] optimized the control of robot grippers using reinforcement learning and camera-based point-cloud data. Specifically, they employed a goal-oriented actor-critic method for 6D gripping, which enabled the robot to effectively grasp objects from various directions and angles. Gupta et al. [18] studied grasping methods using soft grippers. They proposed an algorithm that uses imitation learning to study human movements, choosing and combining a practical selection of these movements. Park et al. [1] conducted research on the easy manufacturing of a four-finger anthropomorphic gripper using a 3D printer. The distinctive feature of this gripper is the application of an impedance controller that allows it to grasp objects of various hardness values.

Research on handovers has been actively pursued, and more recently, studies applying reinforcement learning to solve problems in this domain have increased [19]. Yang et al. [10] placed significant emphasis on the visual aspect by implementing a system that considers various object positions and orientations. They combined closed-loop motion planning with real-time temporal stability in the grasp pose, enabling smooth object grasping. However, the grasping of long objects that generate torque is limited due to the two-finger configuration. Kshirsagar et al. [20] conducted research on training robot controllers using guided policy search (GPS) with the advantage of considering the changes in mass induced by handing over an object. Nevertheless, the study has the limitation of validating the results through simulations without conducting real-world environmental tests. Chang et al. [21] enhanced handover stability through reinforcement learning and end-to-end networks. They used RGB-D images as inputs and were characterized by their applicability even when the object was occluded. However, this study was limited to the robot-to-human handover process, which requires humans to adapt to a robot’s actions. Yang et al. [22] conducted a study using a two-stage learning framework, first training the model while the human remains still and then fine-tuning it in an environment where both the human and the robot move simultaneously. Kedia et al. [23] used a Transformer model with a seven-degree-of-freedom robotic arm, attempting to predict human actions and ignore dependencies. However, this approach has limitations in stability due to its reliance on guessing human behavior. Duan et al. [24] separated grasp prediction and optimization instead of end-to-end learning, sharing similarities with this study in using an anthropomorphic gripper. However, their approach requires separate adjustments for each module and has less adaptability to unexpected obstacles than end-to-end methods. Christen et al. [25] generated large-scale human motion data for robot learning, showing good performance in adapting to objects not present during training, but their approach lacks flexibility in approaching objects from various directions due to optimization focused on a specific grasp direction. This paper uses an anthropomorphic gripper to overcome the stability limitations of a two-finger gripper and employs an end-to-end learning approach to enhance flexibility regarding the direction of objects handed over by humans.

## 3. Framework Setting

In this section, we explain the simulation environment and real-world setup for manipulator-based reinforcement learning of handovers. First, we outline the states for the agent and environment and then elaborate on the structure of both the robot and task environments. Before delving into the specifics of constructing a learning environment, it is essential to outline the three fundamental conditions assumed in this study that must be satisfied to enable robots to receive objects handed by humans naturally.The robot’s gripper must grasp the object where the human placed it to prevent any harm to the human;The robot should be capable of visually distinguishing the object that the human is offering and determining the object’s coordinates;Regardless of the various poses in which the human presents the object, the robot should be able to successfully grasp it.

Figure 1 shows a case that satisfies the three fundamental conditions mentioned above. To satisfy the first condition, we use a camera and YOLO to recognize the object and set the grasping point in the opposite direction to where the person is holding it. Next, to fulfill the second condition, we use an object tracker to transform the relative coordinates of the camera and object based on the reference frame to calculate the location of the manipulator’s end effector. Finally, since the person may not always present the object at the same coordinates, reinforcement learning is employed to allow the manipulator to grasp objects in random poses.

Figure 2 illustrates the simulation training process based on the aforementioned conditions and the steps for implementation in a real environment. During the simulation training, the state of the robot is obtained from the simulation, and the target object is randomly spawned to generate a random grasping point. These data are used in the PPO algorithm for reinforcement learning, and the action derived from the trained network is sent to the simulation controller, enabling the robot to assess the situation in real-time and make appropriate movements.

In the real environment, the ROS framework is applied to obtain the robot’s state. Unlike in the simulation, the target object is recognized, and its location information is obtained by applying YOLO and Darknet ROS. The acquired information is then fed into the trained network to generate an appropriate action, which is used to command the robot’s motors.

### 3.1. Agent and Environment State Setting

In this study, the agent is a manipulator equipped with an anthropomorphic gripper. The HRI hand gripper can grasp various objects, irrespective of their stiffness and shape, through impedance control. However, as shown in Figure 3, it is crucial to position the object at the grip point, which is approximately 3–4 cm from the first joint where the fingers begin. If the gripper alone is unable to move to the object location, the robotic arm performs this task. A UR3 robotic arm is used for this purpose. The agent provides state information by returning the positional $(q_{dof}\in R^{6}$) and velocity (${\dot{q}}_{dof}\in R^{6})$ data of each joint of the UR3, collision status of the links $\left(c_{link}\in R^{1}\right)$, collision status of the HRI_hand $\left(c_{hand}\in R^{1}\right)$, grip point contact with the object $\left(g\in R^{1}\right)$, and quaternion coordinates of the object ${(p}_{t}\in R^{7})$. Action space is the desired joint position ${(a}_{t}\in R^{6})$. Additionally, the spawn location of the object, which has the most significant influence on the target grasping point, is randomly determined within the UR3’s workspace, excluding areas that are too close or beyond the robot’s range. This approach helps to avoid the robot’s singularity. The object’s pose is randomized in the roll direction during training, reflecting the scenario where a person hands over the object at different angles each time.
(1)$$state=\left[q_{dof},{\dot{q}}_{dof},c_{link},c_{hand},g,p_{t}\right]$$
(2)$$action=\left[a_{t}\right]$$

### 3.2. Robot Environment

The robot environment plays a crucial role in facilitating seamless data exchange between simulated or real robots and a reinforcement learning environment. This involves setting up spaces and functionalities where learning takes place. This section describes the configuration of the learning simulation environment.

Reinforcement learning involves thousands to tens of thousands of trial-and-error iterations to obtain results. Therefore, the use of simulations for training robots is efficient. Gazebo is one of the most widely used simulation tools in robotics research [26]. This study uses an open dynamic engine (ODE) in Gazebo to train the agent in an environment that closely mimics reality. Additionally, Gazebo allows for the integration of different sensor plugins such as cameras, contact sensors, LiDAR, and point clouds. This feature enables users to obtain information about the environment and robot movements. The acquired information is then stored in a list format within the state used for training purposes.

In this study, an Intel Realsense camera is attached to the robot for object recognition, and positional information is obtained. YOLO, the object_tracker package, and Darknet ROS are used for this purpose. The rationale behind choosing the RealSense camera lies in its RGB-Depth capabilities, which allow not only image information but also distance data to be captured. Knowing the distance enables the calculation of the position of the object in the x-, y-, and z-axes relative to the camera.

As shown in Figure 4, when the camera captures an image, it is sent to Darknet ROS via a ROS topic, where Darknet classifies the image. Then, YOLO identifies the target object, and the object tracker calculates the coordinates of the object, which are transmitted to the robot server. The robot server plays a crucial role in transmitting information received through ROS topics to the reinforcement learning algorithm. These processes are illustrated in Figure 5.

To facilitate the training of the manipulator in grasping randomly positioned objects, the simulation environment must be configured to randomly spawn objects. This setting is managed within the robot environment, where information about the object’s position, orientation, and other location-related details is encapsulated in a topic and transmitted from the robot server to the Gazebo simulation.

The position of the object should be randomly spawned within the operational workspace of the UR robot and within the field of view of the camera, both of which are components of the agent. Therefore, a specific area was designated for this purpose. This specific area corresponds to the range of 17/12 π ≤ ϕ ≤ 19/12 π in the x-y plane and 0 ≤ θ ≤ 1/6 π in the x-z plane, and the coordinates can be found in Figure 6.

To conduct reinforcement learning, various frameworks are required not only for the Gazebo simulation but also to operate the learning algorithms. Reinforcement learning is predominantly based on the OpenAI Gym library, created by OpenAI. To operate this library, the installation of PyTorch and CUDA is essential. The configured settings for the learning environment and hardware specifications used in this study are listed in Table 1.

### 3.3. Task Environment

The task environment consists of elements that define the characteristics of the environment, with a primary focus on the state, action, and Reward components. In this section, the emphasis is primarily on the configuration of the reward function, which determines the learning direction of the agent. To construct the reward, information about the agent and the surrounding environment needs to be updated, and the agent’s action for each state needs to be determined. Therefore, the state gets updated in real-time, including not only the agent’s state but also information about the target object. This information is reflected in the reward function, which is then updated. The agent receives instructions from the policy on how to proceed with the next action. The reward function is structured around three main terms, as expressed in Equation (3). These terms are updated in real-time at each step and added to the term. Each term has a weight, which allows the robot to prioritize certain aspects. The first term uses the 3D Euclidean distance between the object and the gripper, guiding the gripper to move closer to the object. The second term is crucial because it considers the anthropomorphic nature of the gripper, ensuring that the thumb of the gripper aligns with the *z*-axis of the object during grasping. The third and final term serves to penalize the robot when it collides with its own arm or when parts other than the palm collide with the object. The following is a detailed explanation of each term.

The first term, Euclidean distance, which is illustrated in Figure 7, indicates the shortest 3D distance between the grip point and the target grasping point. The grip point, referring to the region on the HRI hand capable of grasping an object, is located where the palm and fingers converge, measuring approximately 3–4 cm. When the distance between the grip point and the object is smaller than the distance threshold, and the grip point touches the surface of the object, $r_{d}=2$ is assigned. The distance threshold is determined based on the red dashed circle in Figure 3, which represents the optimal gripping range of the HRI hand. The red dash circle has a radius of 3 cm with a margin of ± 3 cm, and the distance threshold is set at 9 cm.
(3)$$r_{b}=w_{d}*r_{d}+w_{o}*r_{o}+w_{c}*r_{c}$$
(4)$$r_{d}=\sqrt{{\left(x_{o}-x_{g}\right)}^{2}+{\left(y_{o}-y_{g}\right)}^{2}+{\left(z_{o}-Z_{g}\right)}^{2}}$$
(5)$$r_{o}=(\left|u_{t\prime z}\right|\left|u_{g\prime y}\right|cos\theta -1)$$
(6)$$r_{c}=\left\{\begin{array}{c} 1\;\;\;\;\;if\;collision\;true\\ \;0\;\;\;\;\;if\;collision\;false \end{array}\right.$$

$w_{d}$: distance weight, $w_{o}$: orientation weight, $w_{c}$: collision weight, $x_{o}$: x-coordinate of object, $y_{o}$: y-coordinate of the object, $z_{o}$: z-coordinate of the object, $x_{g}$: x-coordinate of the grip point, $y_{g}$: y-coordinate of the grip point, $z_{g}$: z-coordinate of the grip point, $u_{t\prime z}$: transformed *z*-axis of the object, $u_{g\prime y}$: transformed *y*-axis of the grip point.

The orientation term, the second part of the reward function, considers the attributes of the HRI hand. Because the HRI hand is designed with human-like characteristics, proper orientation is essential for efficiently gripping an object. Specifically, positioning the thumb finger in alignment with the *z*-axis of an elongated object is recommended for optimal grasping. Therefore, an orientation term is introduced to numerically assess this aspect.

As shown in Figure 8, the coordinate system of the gripper has the *y*-axis pointing in the direction of the thumb finger. Therefore, when the *z*-axis of the object aligns with the *y*-axis of the gripper, the orientation is considered optimal for grasping the object. Transformation matrices are employed to numerically express whether the *y*-axis of the gripper and the *z*-axis of the object are aligned. The base of the UR robot is defined as the reference frame, the frame of the gripper is denoted as {g}, and the frame of the object is denoted as {t}. These two frames are translated so that their origins align with the origin of the reference frame while their directions remain unchanged. We denote the translated frames as {g’} and {t’}, respectively. The rotation matrices for these frames with respect to the reference frame can be expressed by Equations (7) and (8).
(7)Rg′r=Xg′⋅XrYg′⋅XrZg′⋅XrXg′⋅YrYg′⋅YrZg′⋅YrXg′⋅ZrYg′⋅ZrZg′⋅Zr
(8)Rt′r=Xt′⋅XrYt′⋅XrZt′⋅XrXt′⋅YrYt′⋅YrZt′⋅YrXt′⋅ZrYt′⋅ZrZt′⋅Zr

The columns of the rotation matrix represent the transformed unit vectors along each coordinate axis of the reference frame. Therefore, by taking the dot product of the components in the second column of the matrix in Equation (7) and the third column of the matrix in Equation (8), we can numerically determine the angle between the *y*-axis of the gripper and the *z*-axis of the object. Because the vectors are defined as unit vectors, the dot product value approaches 1 as they become more parallel and approaches 0 as they become more perpendicular. For this reason, the second term of the reward function is defined as orientation −1. The subtraction of 1 is intended to reduce the magnitude of the value reflected in the reward. This adjustment is aimed at increasing the influence on the distance when the *y*-axis of the gripper aligns with the *z*-axis of the object. As a result, the reward is structured to guide the gripper closer to the object in the correct direction and orientation. The orientation weight is assigned a positive value to highlight this particular aspect, encouraging the gripper to approach the target grasping point correctly.

The last term of the reward is the collision term. When the robot experiences self-collision or a part other than the grip point touches the object, $r_{c}=-1$ is assigned, and the episode is terminated. If the grip point reaches the object’s grasping point, the episode is also terminated, and a new episode begins.

## 4. Experiments and Evaluation

In this section, we confirm if the orientation factor of the reward, as defined earlier, is accurately represented and evaluate the success rates based on the object’s angle in the built reinforcement learning setting, as illustrated in Figure 9. Furthermore, we confirm the efficacy of the simulation results when implemented on a real robot. Learning and experiments were conducted as follows: Initially, the object is generated in a random position, the previously mentioned reward is then implemented, and learning takes place through the use of the PPO algorithm. In Section 4.1, experiments are conducted to confirm how the orientation term of the reward affects the robot arm’s posture learning by comparing scenarios with and without it. Tests are also carried out with objects placed at random angles of 0, 30, 60, and 90 degrees to assess success rates based on the tilt, and ultimately, the success rate for objects positioned at random angles is confirmed. In Section 4.2, the results obtained from the simulation are applied to a real robot to verify whether it functions correctly.

### 4.1. Simulation

It is important to incorporate a term in the reward that can evaluate orientation when an anthropomorphic gripper grasps an object, as the approach direction plays a crucial role. Consequently, a study was carried out to examine how the proposed orientation term affects learning in the design of the reward function because the approach direction is crucial when an anthropomorphic gripper grasps an object, and it is essential to include a term in the reward that can assess orientation. The training was conducted under the same conditions for all other terms of the reward function, with scenarios divided into cases with or without the orientation term.

Figure 10 and Figure 11 show the robot’s movements with and without the orientation reward term described in Section 3. First, the training results in Figure 10 demonstrate that when the reward function only includes the distance term, the direction of the gripper is not taken into account, resulting in the anthropomorphic gripper colliding with the object using the back of its hand. This occurs because the gripper focuses solely on minimizing the distance to the object. In contrast, Figure 11 shows that with the orientation term included in the reward function, the gripper approaches the object with its palm in a manner that enables it to grasp the object. This is because the orientation term encourages the anthropomorphic gripper’s thumb (aligned with the *y*-axis) to align with the object’s *z*-axis.

When training in scenarios with a reward function based on orientation, it is crucial to incorporate both distance and orientation into the reward calculation. This strategic inclusion is key for aligning the thumb of the HRI hand with the *z*-axis of the target object, which is a critical factor in achieving successful grasping. The effectiveness of this approach is evident in Figure 13, where the reward value steadily increases and eventually converges to approximately 140 after 800,000 steps. This pattern of convergence is a clear indicator of effective learning and adaptation of the system to the task requirements.

Moreover, a reward graph provides valuable insights into the training procedure. As seen in Figure 12, the graph’s slope rises more gradually compared to Figure 13. This is because the back of the gripper’s hand or parts of the manipulator touch the object, leading to less efficient reward increases. When the orientation term is added, it can be observed that the slope of the reward increase becomes steeper as the gripper’s grasping area comes into contact with the object.

The next experiment aimed to assess the success rate of the learned results based on the object’s inclination. The object’s orientation is 0°, classified into two categories: upright and 90° flat. The object is randomly spawned within the space defined in Section 3. The experiment involves tilting the object in four different scenarios, 0°, 30°, 60°, and 90°, testing each scenario 100 times.

Figure 14 illustrates the experimental outcomes based on the inclination angle of the object, with different symbols on the graph representing the position of the gripper at the conclusion of each episode. The blue dots denote successful episodes in which the gripper reached a position appropriate for grasping the object. In contrast, red “x” marks are used for episodes where the gripper makes contact with the object but collides with a part other than the intended grip point. The black “+” marks signify episodes where the gripper failed to make contact with the object, ending the episode in mid-air.

The data in Figure 15 provide an in-depth look at the success rates correlated with various angles of the object. The success rate peaks at 81% when the object is upright at 0°. However, as the tilt angle of the object increased to 30°, the success rate declined slightly to 77% and then more markedly to 54% at a 60° tilt. At a complete tilt of 90°, most episodes resulted in collisions with no successful grasping attempts recorded. An interesting aspect is the aggregate success rate of 63% when the gripper attempted to grasp objects at random angles ranging from 0° to 90°

### 4.2. Real World Experiment

In this section, we present an experiment that applies the learning results to an actual robot to verify its functionality in a real-world environment. The process of handover in a real environment is as follows: when an object is handed over within the view of the RealSense camera, the computer connected to the camera uses YOLO and Object Tracker to identify the object and calculate the coordinates of the grasping point based on the robot’s reference frame (see Figure 16). The laptop controlling the UR3 receives the calculated position information transmitted via a ROS topic. The policy uses this updated information and the robot’s state to generate real-time actions. These actions are sent to the UR3 controller, ensuring that the HRI gripper reaches the correct position to grasp the object.

The results of this experiment are presented in Figure 17 and Figure 18, which show the process unfolding sequentially through steps (a), (b), (c), and (d). We selected a bottle and a banana, considering the weight and size that the HRI gripper can handle, from objects that YOLO can recognize. Both objects were successfully recognized when handed over by a human, and target coordinates were extracted. On average, it took around 15 s for the robot to grasp the object after recognition.

## 5. Conclusions

This study aimed to enable a manipulator equipped with an anthropomorphic gripper to receive various objects through reinforcement learning. Two main environments, the robot environment and the task environment, were established for this purpose. A simulation was set up in the robot environment so that the robot could identify objects and receive coordinate data, with sensors collecting state information as it learned. A reward function was created and implemented in the task environment to guide reinforcement learning, specifying the state and actions of the robot and its environment.

Because the reinforcement-learning environment was implemented on Gazebo, it was easy to replace the manipulator or gripper URDF with different types. This approach can overcome the drawbacks of traditional methods that require new model analysis and controller designs whenever a new robot is configured. Considering that meaningful learning results for a complex 6-axis manipulator can be obtained after approximately 8–12 h of training, this method offers a timesaving advantage. The time needed for learning can be decreased by adjusting the time acceleration in the Gazebo simulation, which was reduced by approximately 2–2.5 times in this study.

In the experiments conducted, as presented in Section 4, to verify whether the learning environment was well-established, it was observed that the robot could move into a position to grasp objects located randomly in various positions. Furthermore, it was proven that the results obtained from the simulation were effective when applied to real robots. However, there was an issue in which collisions occurred, or the gripper failed to reach the graspable position when the object’s inclination increased. This could be due to the increasing kinematic difficulty of the robot in achieving an ideal posture as the object tilts. When the object tilts, the *z*-axis rests horizontally, so the gripper’s *y*-axis must also align horizontally to match the orientation. Human wrists have a high degree of freedom, allowing for easy position changes with minimal movement. However, due to the spacing between the axes, even slight rotations in a manipulator can cause significant errors at the end effector. Although human wrists can move in a wide range of directions with minimal effort, the robot arm can experience significant errors in its end effector position from small rotations because of the long links between joints. This implies that the movement of the manipulator must change significantly when the inclination of the object exceeds a certain degree. However, the PPO algorithm tends to update its policy stably without significant changes, which is a limitation in this context. Furthermore, although a contact sensor located on the Grip point within the simulation enables precise state evaluation, the absence of this sensor data on the real robot resulted in performance disparities compared to the simulation.

Future research should focus on synthesizing an actual robot using simulations. The study also identified the need for additional experiments to determine whether the varying success rates depending on the inclination were due to kinematically challenging poses or the tendency of the PPO algorithm to not pursue significant policy updates. Therefore, comparing the results of applying algorithms other than the PPO and upgrading an actual robot using tactile sensors could lead to enhanced outcomes.

## Figures and Tables

**Figure 1 sensors-24-06275-f001:**
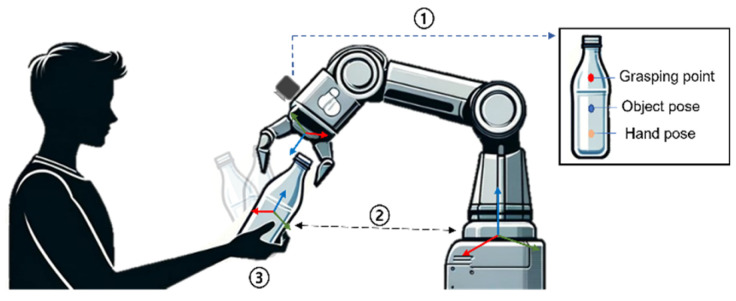
Factors to consider when a robot receives an object.

**Figure 2 sensors-24-06275-f002:**
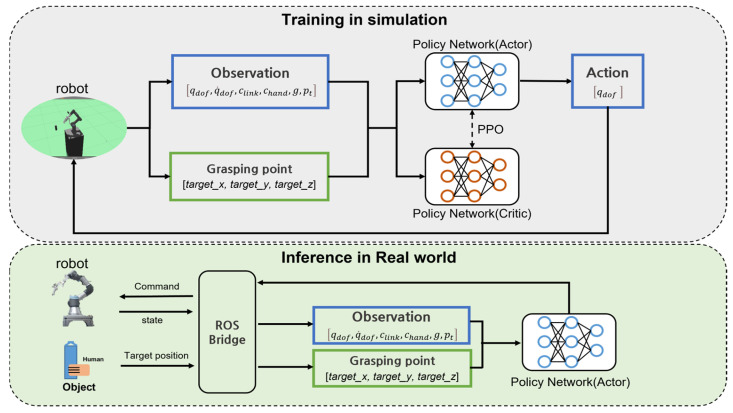
Overview of training and sim-to-real process.

**Figure 3 sensors-24-06275-f003:**
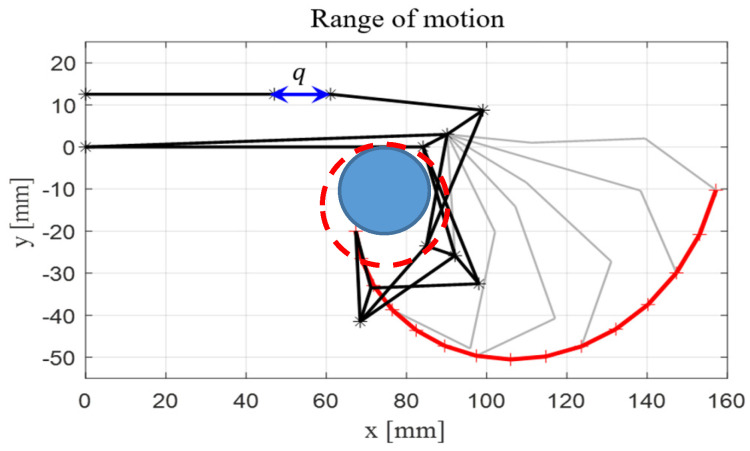
Range of an HRI hand [1]. The dotted line represents the maximum size of the object, and the blue line represents the minimum size of the object.

**Figure 4 sensors-24-06275-f004:**
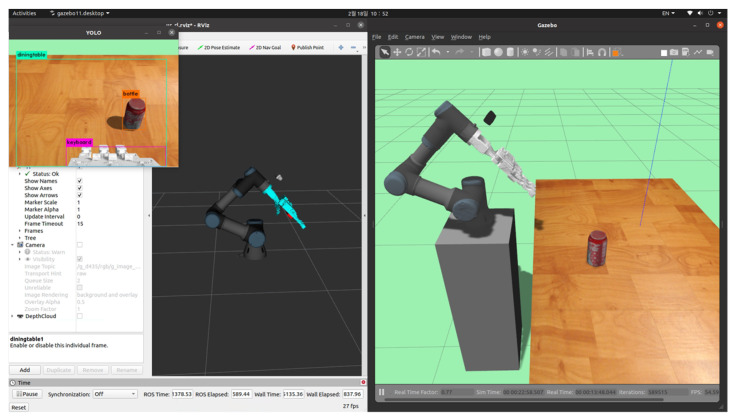
Implementation of YOLO in a Gazebo simulation.

**Figure 5 sensors-24-06275-f005:**
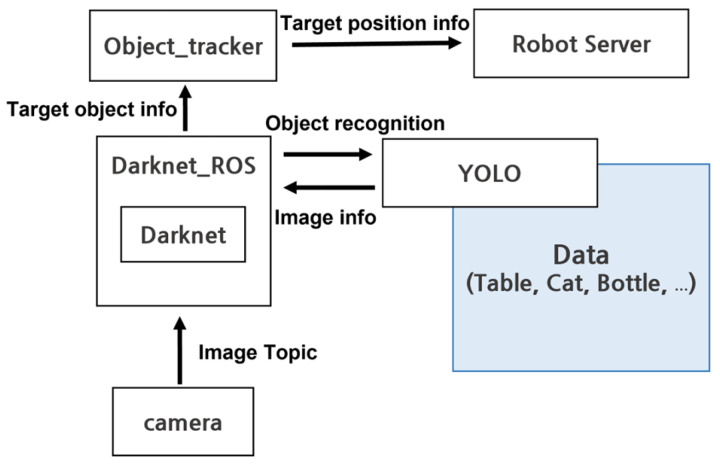
Object recognition and coordinate information acquisition process.

**Figure 6 sensors-24-06275-f006:**
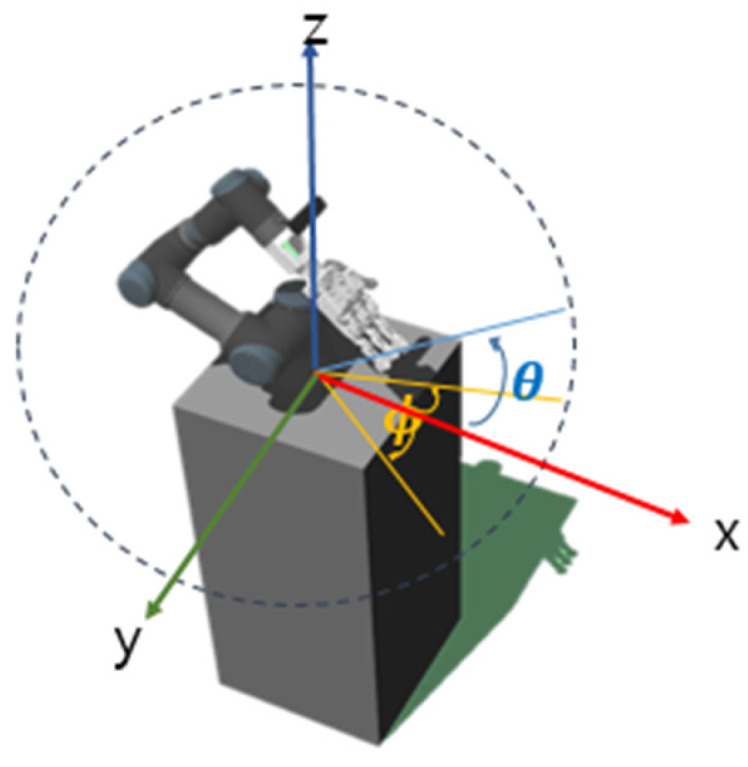
UR robot’s base as the reference coordinate system.

**Figure 7 sensors-24-06275-f007:**
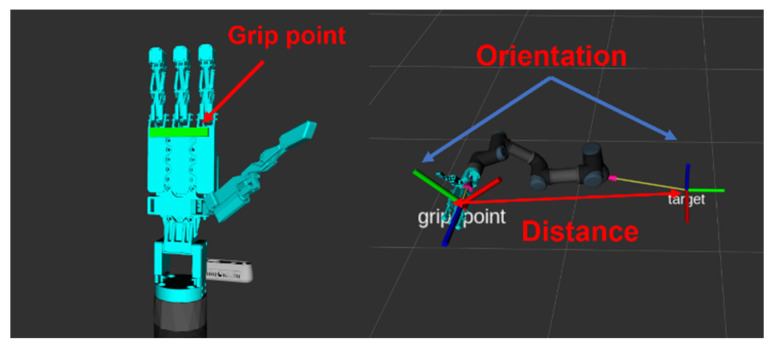
Setting distance and orientation.

**Figure 8 sensors-24-06275-f008:**
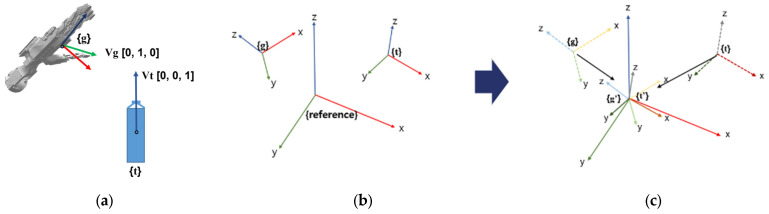
(**a**) Coordinates of the HRI hand and the target. (**b**) Coordinates of the hand and target when the base frame of the manipulator is set as the reference frame. (**c**) Transforming the two coordinates to the origin of the reference frame using a transformation matrix.

**Figure 9 sensors-24-06275-f009:**
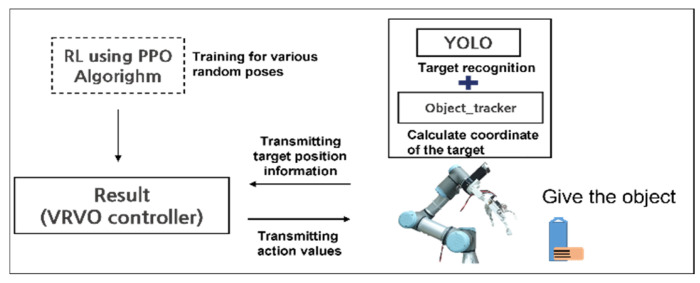
Process of handover after reinforcement learning.

**Figure 10 sensors-24-06275-f010:**
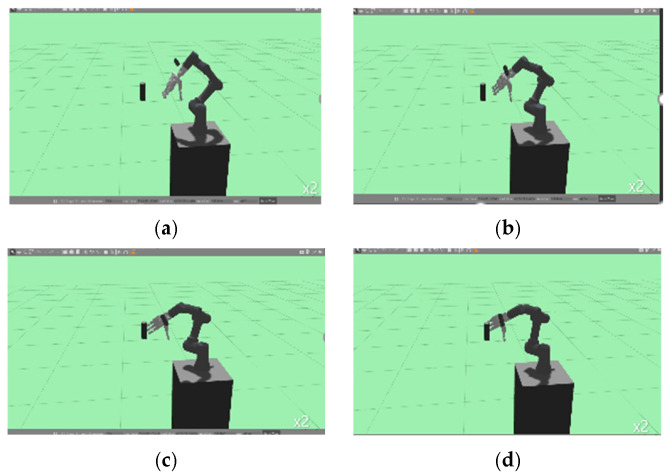
Movement of the UR3 when trained without the orientation term in the reward function. It can be observed that the robot’s thumb moves sequentially from (**a**–**d**), and in (**d**), the thumb points downward, making contact with the object using the back of the hand.

**Figure 11 sensors-24-06275-f011:**
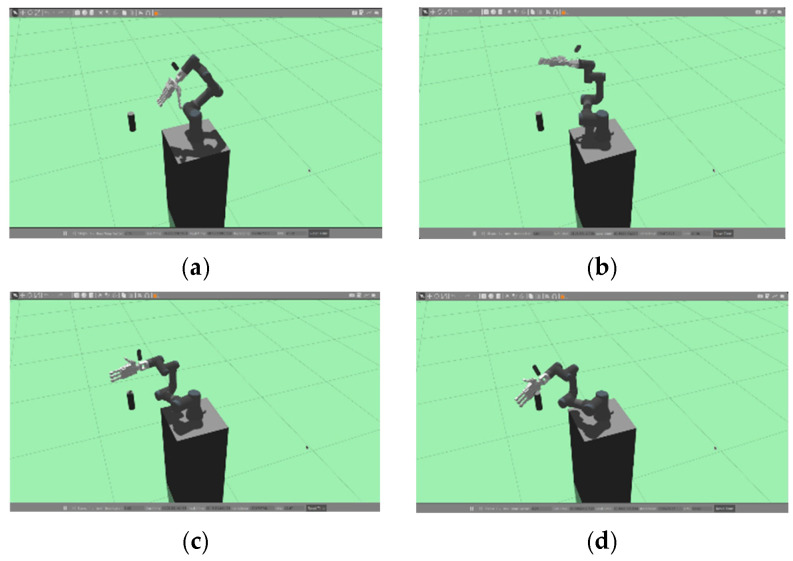
Movement of the UR3 when trained with the orientation term in the reward function. It can be observed that the robot’s thumb moves sequentially from (**a**–**d**), and in (**d**), the thumb points upward, making contact with the object using the palm of the hand.

**Figure 12 sensors-24-06275-f012:**
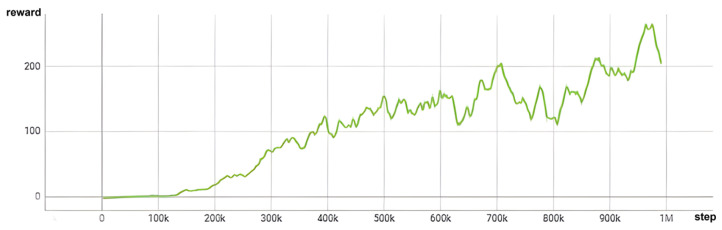
Reward graph without orientation term.

**Figure 13 sensors-24-06275-f013:**
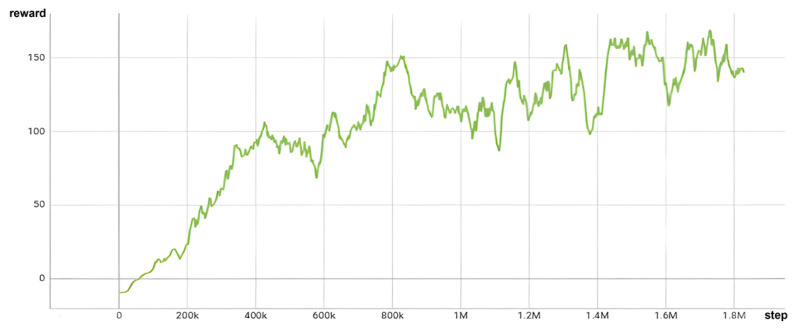
Reward graph with orientation term.

**Figure 14 sensors-24-06275-f014:**
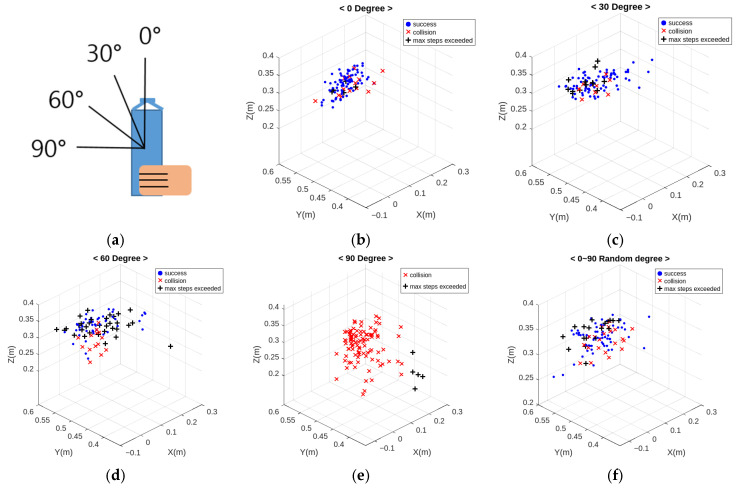
Final position of the grip point according to the inclination of the object. (**a**) shows the degree of tilt of the object handed to the robot, (**b**) shows the result when the object is not tilted, (**c**) shows the result when it is tilted at 30 degrees, (**d**) shows the result at a 60-degree tilt, (**e**) shows the result at a 90-degree tilt, and (**f**) shows the result when the object is randomly tilted between 0 and 90 degrees.

**Figure 15 sensors-24-06275-f015:**
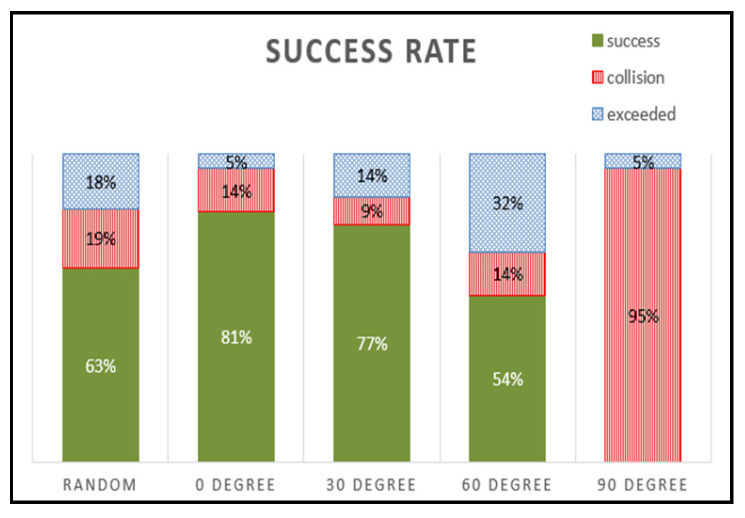
Success rate graph based on object angle.

**Figure 16 sensors-24-06275-f016:**
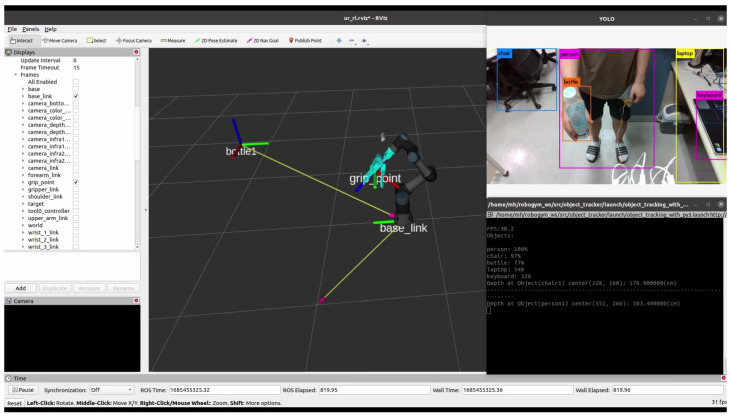
Appearance of recognizing and displaying the position when an object is handed over.

**Figure 17 sensors-24-06275-f017:**
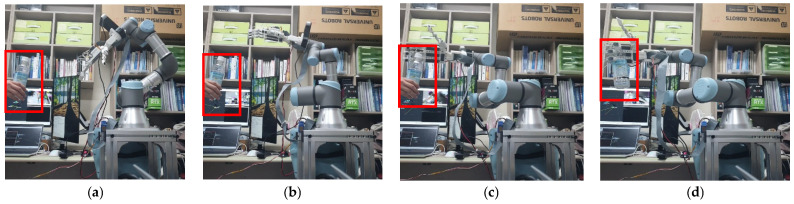
Appearance of applying the learning results to a real robot (target object: bottle). (**a**) shows a person handing over a bottle, and (**b**–**d**) sequentially show the robot recognizing and grasping the object.

**Figure 18 sensors-24-06275-f018:**
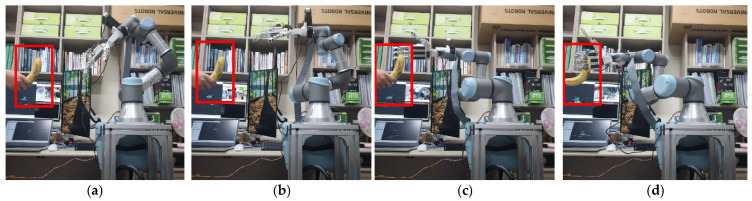
Appearance of applying the learning results to a real robot (target object: banana). (**a**) shows a person handing over a banana, and (**b**–**d**) sequentially show the robot recognizing and grasping the object.

**Table 1 sensors-24-06275-t001:** System specifications used for reinforcement learning.

Feature	Version
ROS version	Noetic
Pytorch	Ver.1.11.0
Python	Ver.3.7.11
RL framework	Robo-gym [27]
Simulation	Gazebo
Learning algorithm	PPO
Graphic card	RTX 3060Ti
CUDA	Ver.11.6
Control rate	125 Hz

## Data Availability

Data are contained within the article.

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
