# Peer review of "Human-to-Robot Handover Based on Reinforcement Learning"

_sensors, 2024, doi:10.3390/s24196275_

Round 1

Reviewer 1 Report

Comments and Suggestions for Authors

The manuscript "Human to Robot handover based on reinforcement learning"  presents a handover scenario between a human and a robot, which is implemented by a reinforcement learning approach.

The paper is interesting but there are several issues which must be addressed

  1. Firstly, the use of reinforcement learning is not sufficiently justified. Which parts of the task need to be learned, and which parts could be managed using traditional control methods? Why didn't the authors consider using some kind of hybrid architecture?

  2. The related work section in the paper is too brief. There are several similar works that should be reviewed and discussed.

  3. The novelty of the paper is unclear. The limitations from the state of the art are not addressed, and the work is not motivated from either a theoretical perspective or practical applications.

  4. Regarding the framework settings, it is necessary to provide an overview of the software architecture. Figure 4 alone is not sufficient for this purpose.

  5. It is unclear whether the three assumptions in Section 3 are prerequisites for the algorithm or if they are managed by the proposed algorithm. If they are assumptions, the authors should clarify that the tests were conducted under these conditions.

  6. How long did the training process take? Which results, if any, stand out as more significant than those in the current state of the art?

  7. The design of the reward function must be described more precisely, as this is the most crucial part of the reinforcement learning framework.

  8. The analysis provided in Section 4.1 is insufficient. Figures 10 and 12 do not clearly demonstrate which reward function is better or why the authors chose a specific one.

Comments on the Quality of English Language

There are no major issues.

Author Response

We thank the reviewer for his/her helpful comments. Below are responses to the particular points raised by the reviewer.

  • The reviewer pointed out that:

“1) The use of reinforcement learning is not sufficiently justified. Which parts of the task need to be learned, and which parts could be managed using traditional control methods? Why didn't the authors consider using some kind of hybrid architecture?”

- (Answer) : As you suggested, we have added reasons for using reinforcement learning to the Introduction. In HRI, an anthropomorphic gripper is more stable for handling various objects than a 2-finger gripper. However, determining the grasping posture becomes more complex compared to using a 2-finger gripper. This complexity makes it challenging to determine the manipulator's posture using traditional methods. Therefore, we used reinforcement learning to reduce modeling time and handle a variety of variables.

Before

After (in the revision)

  • The reviewer pointed out that:

“2)  The related work section in the paper is too brief. There are several similar works that should be reviewed and discussed ”

- (Answer) : We have added studies utilizing reinforcement learning in human-to-robot (H2R) interactions to the Related Work section and provided more detailed descriptions of the papers 

Before

After (in the revision)

  • The reviewer pointed out that:

“3)  The novelty of the paper is unclear. The limitations from the state of the art are not addressed, and the work is not motivated from either a theoretical perspective or practical applications ”

- (Answer) : We have enhanced the Introduction and Related Work sections by adding content regarding the novelty of our research and the limitations of previous studies, as you suggested.

  • The reviewer pointed out that:

“4) Regarding the framework settings, it is necessary to provide an overview of the software architecture. Figure 4 alone is not sufficient for this purpose.”

- (Answer) : As per your comment, a flowchart has been created in Section 3, Framework Setting, to aid understanding, and an explanation has been added accordingly.

Before

After (in the revision)

No flowchart

  • The reviewer pointed out that:

“5)  It is unclear whether the three assumptions in Section 3 are prerequisites for the algorithm or if they are managed by the proposed algorithm. If they are assumptions, the authors should clarify that the tests were conducted under these conditions.”

- (Answer) : The three assumptions mentioned are objectives that our algorithm must achieve for a successful human-to-robot handover. We have clearly described how each objective is implemented in this study.

Before

After (in the revision)

  • The reviewer pointed out that:

“6) How long did the training process take? Which results, if any, stand out as more significant than those in the current state of the art? ”

- (Answer) : The training process took 8 hours in simulation, which we believe is efficient compared to the time required for re-modeling whenever the manipulator or gripper changes in traditional control methods. Additionally, while the latest technologies focus on grasping with a 2-finger gripper, our study demonstrates the ability to grasp objects with an anthropomorphic gripper, which is more challenging in terms of achieving a suitable grasping posture.

  • The reviewer pointed out that:

“7) The design of the reward function must be described more precisely, as this is the most crucial part of the reinforcement learning framework.”

- (Answer) : We are well aware of the point you mentioned. Therefore, we have dedicated approximately two and a half pages to explaining the reward terms. We believe we have provided a detailed explanation, but if there is any specific area that needs further elaboration, please let us know, and we will make the necessary adjustments.

  • The reviewer pointed out that:

“8) The analysis provided in Section 4.1 is insufficient. Figures 10 and 12 do not clearly demonstrate which reward function is better or why the authors chose a specific one..”

- (Answer) : We have revised the interpretation of the reward graph based on your feedback.

Before

After (in the revision)

Thank you for taking the time to provide your feedback.

Reviewer 2 Report

Comments and Suggestions for Authors

This paper is very meaningful in the research of human-computer interaction. Based on reinforcement learning, object transfer between robots and humans has strong application scenarios in human-computer collaboration. This paper can be accepted with some modifications. There are several issues that need to be discussed with the author:

(1) In the simulation environment, the author's modeling only includes objects without human hands, which is inconsistent with the actual robot operating environment? How to achieve stable skill transfer?

(2) How to solve the problem of obstruction during the process of grasping objects with human hands?

(3) Did the author have too few objects during the experiment? How can the experimental results prove robustness?

(4) In the process of human-machine object transmission, human instability should also be a factor that must be considered. The author only discussed it in the conclusion section. Can the author add some object shaking, rotation, or displacement during the experimental stage?

(5) There is also a security issue of human-machine interaction during the process of human-machine transmission, but this issue has not been considered in the virtual environment? Is this somewhat inconsistent with the problem described earlier?

Author Response

We thank the reviewer for his/her helpful comments. Below are responses to the particular points raised by the reviewer.

  • The reviewer pointed out that:

“1) In the simulation environment, the author's modeling only includes objects without human hands, which is inconsistent with the actual robot operating environment? How to achieve stable skill transfer?”

- (Answer) :  In this study, we have achieved the recognition of objects through images and calculated distances to the point where the robot can grasp them. However, as you pointed out, there remains the issue that the robot's grasping position must vary depending on the position of the human hand. When setting up the simulation environment, we randomly selected a point assumed to be where the human hand has grasped, and we set the ideal grasp point for the robot to be located on the opposite side, based on the center of the object. However, when applying this in a real-world environment, it is necessary to recognize the human hand and distinguish it from the object. Since distinguishing the human fingers in contact with the object requires additional research in the field of recognition, it was not covered in this paper and will be addressed in a follow-up study.

  • The reviewer pointed out that:

“2) How to solve the problem of obstruction during the process of grasping objects with human hands? ”

- (Answer) I believe that the problem of passing objects between a human hand and a robot hand in the presence of obstacles is highly challenging. One approach, based on this paper, could involve generating random-sized objects between the robot hand and the target object during simulation training. However, since this method may result in situations where the robot cannot reach the object beyond the obstacles, it would require a deep understanding of the manipulator's workspace.

  • The reviewer pointed out that:

“3) Did the author have too few objects during the experiment? How can the experimental results prove robustness?”

- (Answer)

It may seem that the number of objects is small. However, I believe that it has been demonstrated through training that the robot hand can reach a position where it can grasp objects. The focus then shifts to how many objects can be recognized. In fact, previous studies on the robot hand used in this paper have shown that it can grasp over ten different types of objects using impedance control.

  • The reviewer pointed out that:

“4)  In the process of human-machine object transmission, human instability should also be a factor that must be considered. The author only discussed it in the conclusion section. Can the author add some object shaking, rotation, or displacement during the experimental stage?”

- (Answer) As you commented, in a real environment, unlike in simulation, various noises occur, which can be critical for movements that require precision. Therefore, during the experimental phase, we randomly assigned the position where the object is handed over, and we also added random noise to the rotation of the object to increase the robustness of the training. To achieve even more robust training, considering the addition of speed to the objects could be a potential approach.

  • The reviewer pointed out that:

“5) There is also a security issue of human-machine interaction during the process of human-machine transmission, but this issue has not been considered in the virtual environment? Is this somewhat inconsistent with the problem described earlier?”

- (Answer)

This issue is similar to the one addressed in Response (1). In the simulation setup, we configured the grasping point to be in a position opposite to the random position of the human hand to avoid the robot hand touching the human hand holding the object. However, there remains the additional challenge of recognizing the hand and distinguishing it from the object, which we have decided to address in a follow-up study.

Thank you for taking the time to provide your comments.

Reviewer 3 Report

Comments and Suggestions for Authors

Dear Author,

Your article presents compelling content on the navigation of robotic systems through vision, which is a highly relevant topic today. While the article demonstrates promising results, I have a few suggestions for improvement:

  • Consider adding a flowchart or diagram in the introduction to visually represent the proposed algorithm, which would enhance the reader’s understanding.
  • Expand the discussion of the state of the art to further justify the research problem and the application of reinforcement learning in this context.
  • Provide specific details about the proposed algorithm, such as how it handles disturbances, manages signals from poorly calibrated sensors, etc. Emphasizing the adaptability of reinforcement learning, especially in comparison to traditional methods like Jacobians, would strengthen the argument for its use.
  • Enhance the quality of the images in the article, as some are too small or difficult to interpret.

Overall, your article contains valuable insights. With a few refinements, it could have an even greater impact. Please see the attached document for further details.

Best regards,

Comments on the Quality of English Language

The article is written in generally good English, with only minor errors in writing, but overall, it is well-executed

Author Response

We thank the reviewer for his/her helpful comments. Below are responses to the particular points raised by the reviewer.

  • The reviewer pointed out that:

“1) Consider adding a flowchart or diagram in the introduction to visually represent the proposed algorithm, which would enhance the reader’s understanding.”

- (Answer) : As per your comment, a flowchart has been created in Section 3, Framework Setting, to aid understanding, and an explanation has been added accordingly.

Before

After (in the revision)

No flowchart

  • The reviewer pointed out that:

“2)  Expand the discussion of the state of the art to further justify the research problem and the application of reinforcement learning in this context. ”

- (Answer) : We have strengthened the Introduction with a more detailed problem definition and additional justification for the use of reinforcement learning, reflecting the points you mentioned.

Before

After (in the revision)

  • The reviewer pointed out that:

“3)  Provide specific details about the proposed algorithm, such as how it handles disturbances, manages signals from poorly calibrated sensors, etc. Emphasizing the adaptability of reinforcement learning, especially in comparison to traditional methods like Jacobians, would strengthen the argument for its use.”

- (Answer) : We have added an explanation regarding the object's position to avoid the manipulator's singularity during the reinforcement learning process, as well as the method of adding noise to the target object's pose to enhance robustness

Before

After (in the revision)

  • The reviewer pointed out that:

“4) Enhance the quality of the images in the article, as some are too small or difficult to interpret.”

- (Answer) I have increased the resolution of Fig. 4 as per your notes, and also improved the resolution of Fig. 10, adding the missing x and y labels.

Before

After (in the revision)

Fig. 4

Fig.10

Fig.4

Fig.10

Lastly, we have also incorporated the minor points mentioned in the comments in the PDF file. Thank you for taking the time to provide your feedback.
